# StainNet: Scaling Self-Supervised Foundation Models on Immunohistochemistry and Special Stains for Computational Pathology

**Jiawen Li**[*,1]           JW-LI24@MAILS.TSINGHUA.EDU.CN

**Jiali Hu**[*,2]           72522269@CITYU-DG.EDU.CN

**Xitong Ling**[*,1]           LINGXT23@MAILS.TSINGHUA.EDU.CN

**Yongqiang Lv**[4]           18146676064@163.COM

**Yuxuan Chen**[1]           CHENYX23@MAILS.TSINGHUA.EDU.CN

**Yizhi Wang**[1]           2022214454WANG@GMAIL.COM

**Tian Guan**[†,1]           GUANTIAN@SZ.TSINGHUA.EDU.CN

**Yifei Liu**[†,3]           78909008@QQ.COM

**Yonghong He**[†,1]           HEYH@SZ.TSINGHUA.EDU.CN

[1] *Tsinghua Shenzhen International Graduate School, Tsinghua University, China*

[2] *Biomedical Engineering Programme, City University of Hong Kong (Dongguan), China*

[3] *Department of Pathology, Affiliated Hospital of Nantong University, Nantong, Jiangsu, China*

[4] *Nantong University, Nantong, Jiangsu, China*

**Editors:** Accepted for publication at MIDL 2026

## Abstract

Foundation models trained with self-supervised learning (SSL) on large-scale histological images have significantly accelerated the development of computational pathology. These models can serve as backbones for region-of-interest (ROI) image analysis or patch-level feature extractors in whole-slide images (WSIs) based on multiple instance learning (MIL). Existing pathology foundation models (PFMs) are typically pre-trained on Hematoxylin-Eosin (H&E) stained pathology images. However, images such as immunohistochemistry (IHC) and special stains are also frequently used in clinical practice. PFMs pre-trained mainly on H&E-stained images may be limited in clinical applications involving these non-H&E images. To address this issue, we propose StainNet, a a collection of self-supervised foundation models specifically trained for IHC and special stains in pathology images based on the vision transformer (ViT) architecture. StainNet contains a ViT-Small and a ViT-Base model, both of which are trained using a self-distillation SSL approach on over 1.4 million patch images extracted from 20,231 publicly available IHC and special staining WSIs in the HISTAI database. To evaluate StainNet models, we conduct experiments on three in-house slide-level IHC classification tasks, three in-house ROI-level special stain and two public ROI-level IHC classification tasks to demonstrate their strong ability. We also perform ablation studies such as few-ratio learning and retrieval evaluations, and compare StainNet models with recent larger PFMs to further highlight their strengths. The StainNet model weights are available at https://github.com/WonderLandxD/StainNet.

**Keywords:** Non-H&E images, foundation model, computational pathology.

---

[*] Contributed equally

[†] Corresponding author

## 1. Introduction

Modern computational pathology involves the use of deep learning models, such as convolutional neural networks (He et al., 2016) and ViT (Dosovitskiy, 2020), for clinical tasks on histopathological images, including cancer screening (Bejnordi et al., 2017), tumor analysis (Lu et al., 2021), morphological retrieval (Chen et al., 2022a), and survival prognosis (Chen et al., 2021a). In recent years, SSL methods have gained attention and have been proven to be highly effective (Kang et al., 2023). These methods design pretext tasks, such as contrastive learning (Chen et al., 2020) and image reconstruction (He et al., 2022), allowing the model to learn effective representations from unlabeled data and obtain good initialization weights, which can then be directly used as foundation models for pathology image feature extraction.

Most existing PFMs are pre-trained on H&E-stained histological images, which are routinely used and easily accessible in clinical workflows. As a result, these PFMs typically perform well with H&E-stained images. For example, by directly analyzing expert-annotated H&E-stained ROIs, PFMs can be used to identify tumor malignancy or microenvironment patterns (Lin et al., 2025). Additionally, when H&E resection or biopsy WSIs are cropped into patches, PFMs can extract these into high-dimensional features, which are used in weakly-supervised learning tasks based on MIL (Xu et al., 2025), such as cancer screening or biomarker detection (Campanella et al., 2025). However, there are other staining in clinical practice. For example, by using antibodies that bind to proteins in tissue cells, IHC slides are widely used to visualize the presence, location, and expression levels of specific proteins, which are commonly utilized for precision cancer diagnosis, disease subtyping, and drug selection. Due to significant differences in nuclear color, tissue patterns, and texture compared to H&E staining, directly transferring existing PFMs to these non-H&E images for representation may be limited in clinical performance (Li et al., 2024b).

To address this issue, we propose StainNet, a collection of PFMs specifically pre-trained on IHC and special stain images. Specifically, to train StainNet, we first select 20,231 non-H&E WSIs from the publicly available large-scale HISTAI database (Nechaev et al., 2025). For each WSI, we randomly crop up to 100 tissue-containing patch images to create our pre-training dataset. Next, to develop StainNet, we utilize DINO (Caron et al., 2021), a lightweight self-supervised learning framework specifically designed for ViT models, enabling StainNet to learn task-agnostic morphological information from a large collection of unlabeled special staining images. Finally, to evaluate StainNet, we construct downstream task datasets at both the ROI and WSI levels and compare textcolorredthe performance with existing pathology foundation models. Through this work, we aim to explore the adaptability of PFMs in clinical tasks involving non-H&E images and call for the broader exploration of computational pathology in more diverse clinical pathology image datasets.

## 2. Related works

### 2.1. Self-supervised PFMs

Foundation models are accelerating the advancement of computational pathology. By performing SSL on large-scale unlabeled pathology images, models can obtain histology-specific weights, which significantly improve their performance on downstream tasks. For instance,

CTransPath (Wang et al., 2022) uses a MoCov3-based SSL approach (Chen et al., 2021b) and is trained on 15.6 million image patches from 25 anatomical sites to demonstrate its ability in H&E staining image retrieval. HIPT (Chen et al., 2022b) and Lunit (Kang et al., 2023) adopt DINO (Caron et al., 2021), validating the feasibility of pre-training ViT models on histological images. Given the easy scalability of ViT parameters, recent works have aimed to train on larger datasets with larger ViT models. Examples include UNI (Chen et al., 2024), Virchow (Vorontsov et al., 2024), and Prov-Gigapath (Xu et al., 2024), which aim to diversify the pre-training dataset with more varied pathology images and leverage more computational resources to build stronger general pathology representation models. Additionally, some studies explore SSL in conjunction with special staining, such as Patho-Duet (Hua et al., 2024), which utilizes registered H&E and IHC images to help the model learn cross-staining features, aiding the transfer to more advanced nuclear information.

### 2.2. IHC and Special staining in clinical pathology

H&E staining is the most commonly used histological staining method, providing basic structural information by staining the cell nuclei and cytoplasm. However, H&E staining does not fully reveal certain cell features or tissue details, especially in the detection of immune responses or protein expression. To address these limitations, several staining methods such as IHC, Masson's trichrome, and acid phosphatase staining are developed to highlight the specific proteins, cell subtypes, and extracellular matrix components (Gridley, 1957). For example, IHC utilizes antibodies to bind to target proteins, making it a valuable tool for cancer diagnosis and disease subtyping (Magaki et al., 2018); Masson's trichrome staining is employed to visualize elastic fibers in tissues, particularly in fibrosis-related research (Lefkowitch, 2006). Recently, the development of computational pathology has accelerated research into the application of computer vision models to these special staining images, such as lymphocyte detection in IHC images (Swiderska-Chadaj et al., 2019) and the use of multiple special stains for kidney biopsy evaluation (Jayapandian et al., 2021). Some studies have also explored the direct use of generative models to convert H&E images into these stains to improve diagnostic accuracy (De Haan et al., 2021; Bai et al., 2023; Pati et al., 2024; Yan et al., 2023). However, these non-H&E images differ significantly from H&E staining in terms of staining mechanisms, texture features, and visual appearance, making it difficult for large-scale pre-trained models based on general H&E staining images to capture the specific features of special staining images (Li et al., 2024b).

## 3. Methodology

We develop and evaluate our StainNet models through data collection, model pre-training, and downstream testing. The entire process is illustrated in Figure 1.

### 3.1. Data collection

We use the publicly available large-scale WSI database HISTAI (Nechaev et al., 2025) as our pre-training data. The original HISTAI dataset contains 112,801 WSIs, of which 20,265 are non-H&E stains. We first apply the OTSU thresholding algorithm (Otsu et al., 1975) to exclude 34 special staining WSIs that can not extract foreground tissue, resulting in

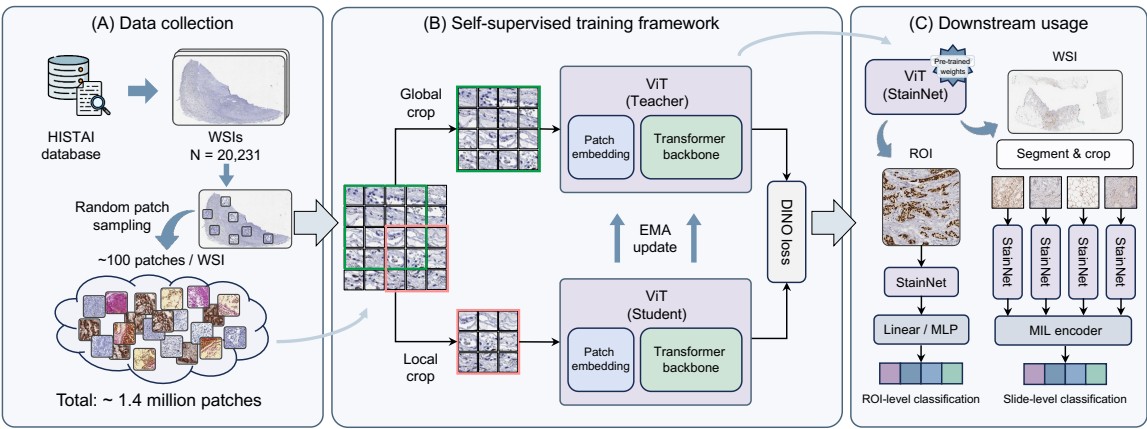

Figure 1: Overview of our proposed StainNet. We first collect approximately 1.4 million patch images with a resolution of $224 \times 224$ pixels from the HISTAI database. Then, we train StainNet models using the DINO SSL method. Finally, we adapt StainNet to downstream ROI and WSI tasks to explore its performance.

20,231 WSIs. Then, for each WSI, we perform non-overlapping $224 \times 224$ pixel sliding window operations on the foreground tissue, and randomly crop 100 image patches (if a WSI contains fewer than 100 patches, all the patches will be selected). In total, we obtain 1,418,938 patches for subsequent StainNet pre-training. More pre-training data details are provided in Appendix A.

### 3.2. Model pre-training

Our proposed StainNet models adopt the SSL method DINO (Caron et al., 2021) as the pre-training strategy to let the ViT model learn robust feature representations from IHC and special stain pathology images without manual annotations. DINO consists of two networks with the same architecture: a student network optimized via gradient descent and a teacher network updated using an exponential moving average (EMA) of the student weights. We follow the original DINO approach for data augmentation and multi-crop strategies , which are provided in Appendix B. Specifically, we generate a set of global and local views for each pathology image, and the student takes all views as input, while the teacher only receives the global views. By minimizing the cross-entropy loss between the output probability distributions of the two networks, the student is encouraged to infer global tissue context solely from local information. In addition, centering and sharpening operations are applied to the teacher outputs to prevent mode collapse. The detailed hyperparameters are provided in the implementation and Appendix B.

### 3.3. Downstream adaptation

Our pre-trained StainNet can be applied to both ROI and WSI analysis for special staining images. For ROI tasks, the extracted features from StainNet can be passed to either a single linear layer (linear probe) or a multilayer perceptron (MLP probe) consisting of two linear

layers with a ReLU($\cdot$) activation function for downstream classification. For WSI tasks, we first crop a set of tissue-containing patch images and then perform offline feature extraction using StainNet. The resulting $N \times C$ feature sequence (where $N$ is the number of image patches and $C$ is the feature dimension) can be fed into a series of MIL models to obtain slide-level representations and classification scores.

## 4. Experiments

### 4.1. Downstream datasets and implementation

We evaluate StainNet on three private IHC WSI tasks, NTUH-*Ki67*-Liver, *P53*-UCEC, and *MLH1*-UCEC, two public available IHC ROI tasks, BCI (Liu et al., 2022) and MIST (Li et al., 2023), and three in-house special stain ROI tasks, Glomerulus-Masson, Glomerulus-PAS, and Glomerulus-PASM, which are derived as subsets from our previous work (He et al., 2025). The NTUH-*Ki67*-Liver 3-class dataset consists of 46 *Ki67* benign liver slides, 43 *Ki67* hepatocellular carcinoma slides, and 50 *Ki67* normal liver slides. The *P53*-UCEC 5-class dataset consists of 139 *P53* wild-type, 38 *P53* mutant, 10 *P53* null, 14 *P53* subclonal, and 17 *P53* endometrial carcinoma slides. The *MLH1*-UCEC 3-class dataset consists of 30 *MLH1* deficient, 155 *MLH1* proficient, and 10 slides with partial loss of *MLH1* expression from endometrial carcinoma. All slides are collected from the Affiliated Hospital of Nantong University and scanned using the SQS-120P scanning system from Shenzhen Shengqiang Technology Co., Ltd. The annotations are performed and verified at the slide level by two pathologists. The BCI dataset is a collection of 4,870 paired H&E-IHC images showing *HER2* biomarker expression. We use the IHC images and categorize them into four *HER2* expression levels: 0, 1+, 2+, and 3+, with 240, 1,153, 2,142, and 1,335 samples, respectively. The MIST dataset is a large-scale H&E-IHC paired image dataset. We use the IHC images corresponding to four biomarkers: 5,642 *HER2*, 5,361 *Ki67*, 5,153 *ER*, and 5,139 *PR* to construct a 4-class dataset for evaluating the ability of PFMs to identify biomarkers across different staining conditions. The Glomerulus-Masson dataset consists of 200 normal, 57 slight, 112 moderate, and 113 severe Masson-stained glomeruli images. The Glomerulus-PAS dataset consists of 1200 normal, 1129 slight, 479 moderate, and 379 severe PAS-stained glomeruli images. The Glomerulus-PASM dataset consists of 200 normal, 76 slight, 135 moderate, and 87 severe PASM-stained glomeruli images.

We further evaluate StainNet models on two public H&E-stained ROI tasks, CRC-100K (Kather et al., 2019a) and KatherMS (Kather et al., 2019b), as well as one WSI task, PANDA-*Karo.* (Bulten et al., 2022), to investigate its capability on H&E image analysis. CRC-100K consists of nine categories: adipose, background, debris, lymphocytes, mucus, smooth muscle, normal colon mucosa, cancer-associated stroma, and colorectal adenocarcinoma epithelium. We perform label-stratified splitting on the official NCT-CRC-HE-100K dataset with a ratio of 0.8:0.2 to construct the train–val set, and use CRC-VAL-HE-7K as the test set. KatherMS contains two categories: microsatellite stable and microsatellite instable. We similarly conduct label-stratified splitting on the official training set with a ratio of 0.8:0.2 to obtain the train–val set, and use the official test set for evaluation. PANDA-*Karo.* consists of 5455 biopsy WSIs from Karolinska Institute, including 1924 G0, 1814 G1, 668 G2, 317 G3, 481 G4, and 251 G5. We label-stratify the train-val-test fold with ISUP grading ground truth of 0.5:0.2:0.3 for experiments.

| Patch encoder | MIL encoder | | | | | | |
| *NTUH-Ki67-Liver* | ABMIL | SiMLP | TransMIL | WiKG | AMDMIL | S4MIL | Overall |
|---|---|---|---|---|---|---|---|
| ResNet-50 (He et al., 2016) | $0.742_{0.093}$ | $0.712_{0.055}$ | $0.829_{0.055}$ | $0.836_{0.059}$ | $0.838_{0.071}$ | $0.832_{0.076}$ | 0.7980 |
| CTransPath (Wang et al., 2022) | $0.668_{0.050}$ | $0.757_{0.105}$ | $0.858_{0.040}$ | $0.860_{0.041}$ | $0.802_{0.018}$ | $0.819_{0.043}$ | 0.7940 |
| PathoDuet (Hua et al., 2024) | $0.655_{0.033}$ | $0.641_{0.024}$ | $0.699_{0.033}$ | $0.738_{0.032}$ | $0.716_{0.062}$ | $0.725_{0.030}$ | 0.6957 |
| HIPT (Chen et al., 2022b) | $0.706_{0.086}$ | $0.753_{0.053}$ | $0.814_{0.050}$ | $0.815_{0.058}$ | $0.835_{0.020}$ | $0.843_{0.017}$ | 0.7943 |
| Lunit (Kang et al., 2023) | $0.747_{0.048}$ | $0.838_{0.041}$ | $0.872_{0.051}$ | $0.891_{0.044}$ | $0.872_{0.020}$ | $0.862_{0.032}$ | 0.8470 |
| UNI (Chen et al., 2024) | $0.764_{0.055}$ | $\underline{0.891}_{0.052}$ | $0.886_{0.036}$ | $0.914_{0.033}$ | $0.908_{0.040}$ | $0.890_{0.052}$ | 0.8755 |
| PathOrchestra (Yan et al., 2025) | $0.689_{0.076}$ | $0.837_{0.068}$ | $0.850_{0.040}$ | $0.861_{0.093}$ | $0.905_{0.040}$ | $0.878_{0.054}$ | 0.8367 |
| GPFM (Ma et al., 2025) | $0.754_{0.060}$ | $0.849_{0.055}$ | $0.899_{0.070}$ | $0.900_{0.051}$ | $0.900_{0.049}$ | $0.876_{0.034}$ | 0.8630 |
| UNI-2 (Chen et al., 2024) | $0.798_{0.049}$ | $0.842_{0.087}$ | $\underline{0.907}_{0.041}$ | $\mathbf{0.922}_{0.017}$ | $\mathbf{0.922}_{0.046}$ | $\underline{0.909}_{0.033}$ | 0.8833 |
| Virchow-2 (Zimmermann et al., 2024) | $0.814_{0.050}$ | $0.884_{0.071}$ | $0.895_{0.067}$ | $0.915_{0.056}$ | $0.907_{0.022}$ | $\mathbf{0.922}_{0.063}$ | $\underline{0.8895}$ |
| Prov-GigaPath (Xu et al., 2024) | $\mathbf{0.852}_{0.098}$ | $0.876_{0.080}$ | $0.884_{0.077}$ | $0.878_{0.054}$ | $\underline{0.913}_{0.030}$ | $0.905_{0.047}$ | 0.8847 |
| H-optimus-0 (Saillard et al., 2024) | $0.779_{0.098}$ | $0.841_{0.057}$ | $0.899_{0.069}$ | $0.859_{0.047}$ | $0.889_{0.046}$ | $0.888_{0.045}$ | 0.8592 |
| **StainNet-Small (Ours)** | $0.801_{0.092}$ | $0.873_{0.042}$ | $0.896_{0.073}$ | $0.916_{0.030}$ | $0.892_{0.051}$ | $0.883_{0.060}$ | 0.8768 |
| **StainNet-Base (Ours)** | $\underline{0.830}_{0.084}$ | $\mathbf{0.892}_{0.025}$ | $\mathbf{0.928}_{0.050}$ | $\underline{0.921}_{0.048}$ | $0.887_{0.028}$ | $0.900_{0.048}$ | **0.8930** |

Table 1: Comparison of balanced accuracy results for our StainNet with 12 PFMs on the NTUH-*Ki67*-Liver classification dataset. We use 6 different MIL methods for slide-level fine-tuning and report the average evaluation metrics with standard deviation across all folds. The best result is in **bold** and the second best is underlined.

For DINO training, we use the AdamW optimizer with a learning rate of $5 \times 10^{-4}$ to pre-train StainNet for 100 epochs with a batch size of 256. For multi-cropping, global crops are sampled using a random resize scale in the range of 0.4-1.0, and 8 local crops are generated with a scale in the range of 0.05-0.4. For EMA design, we increase teacher momentum linearly from 0.9995 to 1.0 throughout training. The weight decay is scheduled from 0.04 to 0.4, and the teacher temperature is set to 0.04 without warm-up and last layer freezing. Our StainNet model family includes StainNet-Small and StainNet-Base, which are initialized with ImageNet-pre-trained (Deng et al., 2009) ViT-Small and ViT-Base backbones, and are then further pre-trained using DINO. We also enable normalization of the last layer to stabilize the training process. The StainNet-Small experiment is conducted on a single NVIDIA RTX PRO 6000 96GB GPU with mixed-precision and takes approximately 3.04 days to complete. The StainNet-Base experiment is conducted on 8 NVIDIA A100 80GB GPU and tasks approximately 2.78 days to complete. For both StainNet-Small and StainNet-Base, we select the last epoch checkpoint as the final model weights.

For downstream fine-tuning, we use AdamW with a learning rate of $10^{-4}$ and a weight decay of $10^{-4}$. We set 20 epochs with a batch size of 1 for the WSI task and 20 epochs with a batch size of 128 for the ROI task. For classification tasks, we save the fine-tuned model weights based on the best balanced accuracy, and also report accuracy, AUC, and F1-score from each fold. Early stopping is applied with a patience of 5 epochs, meaning that training is terminated if the balanced accuracy on the validation set does not improve for five consecutive epochs. For retrieval tasks, we perform image matching using cosine similarity and report Recall@K and mAP@K, where K ranges from 1 to 20. For NTUH-*Ki67*-Liver, *P53*-UCEC, *MLH1*-UCEC, BCI, and MIST, we adopt a label-stratified 5-fold

| Patch encoder | P53-UCEC | | | | MLH1-UCEC | | | |
|---|---|---|---|---|---|---|---|---|
| | Acc. | Bal acc. | AUC | F1-score | Acc. | Bal acc. | AUC | F1-score |
| ResNet-50 (He et al., 2016) | $0.795_{0.000}$ | $0.333_{0.000}$ | $0.602_{0.147}$ | $0.704_{0.000}$ | $0.638_{0.008}$ | $0.200_{0.000}$ | $0.618_{0.057}$ | $0.497_{0.011}$ |
| CTransPath (Wang et al., 2022) | $0.795_{0.000}$ | $0.333_{0.000}$ | $0.705_{0.061}$ | $0.704_{0.000}$ | $0.785_{0.024}$ | $0.376_{0.019}$ | $0.779_{0.094}$ | $0.705_{0.026}$ |
| PathoDuet (Hua et al., 2024) | $0.795_{0.000}$ | $0.333_{0.000}$ | $0.362_{0.153}$ | $0.704_{0.000}$ | $0.762_{0.041}$ | $0.361_{0.029}$ | $0.689_{0.087}$ | $0.688_{0.035}$ |
| HIPT (Chen et al., 2022b) | $0.795_{0.000}$ | $0.333_{0.000}$ | $0.671_{0.101}$ | $0.704_{0.000}$ | $0.706_{0.069}$ | $0.285_{0.080}$ | $0.592_{0.185}$ | $0.600_{0.102}$ |
| Lunit (Kang et al., 2023) | $0.877_{0.056}$ | $0.538_{0.117}$ | $0.740_{0.133}$ | $0.840_{0.081}$ | $0.775_{0.018}$ | $0.369_{0.026}$ | $0.745_{0.044}$ | $0.695_{0.014}$ |
| UNI (Chen et al., 2024) | $\underline{0.933}_{0.023}$ | $\underline{0.642}_{0.049}$ | $0.833_{0.064}$ | $\underline{0.908}_{0.024}$ | $\mathbf{0.803}_{0.012}$ | $0.393_{0.012}$ | $\underline{0.853}_{0.024}$ | $0.722_{0.017}$ |
| PathOrchestra (Yan et al., 2025) | $0.913_{0.014}$ | $0.616_{0.039}$ | $0.852_{0.042}$ | $0.889_{0.015}$ | $0.761_{0.027}$ | $0.367_{0.046}$ | $0.757_{0.057}$ | $0.684_{0.034}$ |
| GPFM (Ma et al., 2025) | $\mathbf{0.933}_{0.014}$ | $\mathbf{0.642}_{0.029}$ | $0.842_{0.081}$ | $\mathbf{0.909}_{0.014}$ | $\mathbf{0.803}_{0.012}$ | $0.393_{0.012}$ | $0.830_{0.047}$ | $0.722_{0.016}$ |
| UNI-2 (Chen et al., 2024) | $0.903_{0.053}$ | $0.567_{0.114}$ | $0.835_{0.139}$ | $0.870_{0.063}$ | $0.780_{0.019}$ | $0.379_{0.020}$ | $0.790_{0.063}$ | $0.700_{0.015}$ |
| Virchow-2 (Zimmermann et al., 2024) | $0.913_{0.034}$ | $0.616_{0.049}$ | $\underline{0.862}_{0.040}$ | $0.888_{0.034}$ | $0.775_{0.023}$ | $0.382_{0.014}$ | $0.815_{0.054}$ | $0.700_{0.016}$ |
| Prov-GigaPath (Xu et al., 2024) | $0.908_{0.029}$ | $0.614_{0.047}$ | $0.826_{0.084}$ | $0.884_{0.028}$ | $0.785_{0.024}$ | $\underline{0.427}_{0.076}$ | $0.850_{0.047}$ | $\underline{0.724}_{0.033}$ |
| H-optimus-0 (Saillard et al., 2024) | $0.903_{0.033}$ | $0.603_{0.058}$ | $0.804_{0.110}$ | $0.878_{0.035}$ | $0.785_{0.011}$ | $0.384_{0.013}$ | $0.808_{0.064}$ | $0.706_{0.012}$ |
| **StainNet-Small (Ours)** | $0.892_{0.059}$ | $0.562_{0.135}$ | $0.706_{0.218}$ | $0.854_{0.087}$ | $0.785_{0.024}$ | $0.373_{0.029}$ | $0.723_{0.032}$ | $0.704_{0.023}$ |
| **StainNet-Base (Ours)** | $\mathbf{0.933}_{0.014}$ | $\mathbf{0.642}_{0.029}$ | $\mathbf{0.880}_{0.068}$ | $\mathbf{0.909}_{0.014}$ | $\underline{0.803}_{0.025}$ | $\mathbf{0.536}_{0.120}$ | $\mathbf{0.874}_{0.064}$ | $\mathbf{0.754}_{0.034}$ |

Table 2: Comparison results for our StainNet with 12 PFMs on the P53-UCEC and MLH1-UCEC classification datasets. We use ABMIL methods for slide-level fine-tuning and report four average evaluation metrics with standard deviation across all folds. The best result is in **bold** and the second best is underlined.

train–val cross-validation setting for evaluation. For Glomerulus-Masson, Glomerulus-PAS, and Glomerulus-PASM, we follow our previous work (Li et al., 2024b) to employ a label-stratified train–val–test split of 0.5:0.2:0.3 with five random seeds. For H&E tasks, we use the train-val-test setting with three random seeds. We use a single NVIDIA RTX 3090 24GB GPU with full-precision computing to run all downstream tasks. For MIL and PFM baselines, we follow their official implementations, including hyperparameter settings and data preprocessing pipelines. Except for these model-specific configurations, all other downstream fine-tuning hyperparameters are kept identical to those used in StainNet.

## 4.2. Comparison results on WSI tasks

For the WSI task, we compare the proposed StainNet-Small and StainNet-Base with 12 PFMs from small to large parameter size: (1) ResNet-50 (ImageNet pre-trained) (He et al., 2016), (2) CTransPath (Wang et al., 2022), (3) PathoDuet (Hua et al., 2024), (4) HIPT (Chen et al., 2022b), (5) Lunit (Kang et al., 2023), (6) UNI (Chen et al., 2024), (7) PathOrchestra (Yan et al., 2025), (8) GPFM (Ma et al., 2025), (9) UNI-2 (Chen et al., 2024), (10) Virchow-2 (Zimmermann et al., 2024), (11) Prov-GigaPath (Xu et al., 2024), and (12) H-optimus-0 (Saillard et al., 2024). For the WSI task, we evaluate six MIL models: (1) ABMIL (Ilse et al., 2018), a gated-attention MIL, (2) SiMLP (Li et al., 2025), a method combining unsupervised average pooling with an MLP classifier, (3) TransMIL (Shao et al., 2021), a transformer-based MIL, (4) WiKG (Li et al., 2024a), a dynamic graph representation method, (5) AMDMIL (Ling et al., 2024), a method combining masked denosing with agent tokens, and (6) S4MIL (Fillioux et al., 2023), a state space model-based MIL. Table 1 shows the comparison results of the StainNet series models on the six MIL models on NTUH-Ki67-Liver.

| Patch encoder | Linear probe | | | | MLP probe | | | |
|---|---|---|---|---|---|---|---|---|
| BCI | Acc. | Bal acc. | AUC | F1-score | Acc. | Bal acc. | AUC | F1-score |
| ResNet-50 (He et al., 2016) | $0.587_{0.010}$ | $0.407_{0.010}$ | $0.774_{0.018}$ | $0.541_{0.014}$ | $0.709_{0.016}$ | $0.614_{0.019}$ | $0.900_{0.008}$ | $0.703_{0.016}$ |
| CTransPath (Wang et al., 2022) | $0.575_{0.011}$ | $0.387_{0.007}$ | $0.808_{0.007}$ | $0.499_{0.008}$ | $0.736_{0.015}$ | $0.653_{0.021}$ | $0.913_{0.008}$ | $0.733_{0.015}$ |
| PathoDuet (Hua et al., 2024) | $0.612_{0.016}$ | $0.430_{0.015}$ | $0.810_{0.014}$ | $0.554_{0.028}$ | $0.687_{0.019}$ | $0.564_{0.044}$ | $0.877_{0.012}$ | $0.664_{0.024}$ |
| HIPT (Chen et al., 2022b) | $0.564_{0.023}$ | $0.404_{0.023}$ | $0.754_{0.020}$ | $0.532_{0.028}$ | $0.661_{0.009}$ | $0.533_{0.010}$ | $0.859_{0.010}$ | $0.650_{0.011}$ |
| Lunit (Kang et al., 2023) | $0.696_{0.015}$ | $0.545_{0.018}$ | $0.880_{0.008}$ | $0.681_{0.017}$ | $0.872_{0.016}$ | $0.816_{0.033}$ | $0.976_{0.002}$ | $0.871_{0.017}$ |
| UNI (Chen et al., 2024) | $0.785_{0.020}$ | $0.693_{0.031}$ | $0.936_{0.011}$ | $0.781_{0.021}$ | $0.900_{0.011}$ | $0.877_{0.024}$ | $0.984_{0.004}$ | $0.900_{0.011}$ |
| PathOrchestra (Yan et al., 2025) | $0.732_{0.021}$ | $0.601_{0.024}$ | $0.914_{0.009}$ | $0.721_{0.022}$ | $0.847_{0.012}$ | $0.804_{0.035}$ | $0.969_{0.004}$ | $0.846_{0.012}$ |
| GPFM (Ma et al., 2025) | $0.704_{0.004}$ | $0.528_{0.005}$ | $0.902_{0.008}$ | $0.681_{0.004}$ | $0.844_{0.017}$ | $0.816_{0.026}$ | $0.966_{0.006}$ | $0.845_{0.017}$ |
| UNI-2 (Chen et al., 2024) | $0.772_{0.010}$ | $0.624_{0.019}$ | $0.934_{0.007}$ | $0.759_{0.012}$ | $0.874_{0.013}$ | $0.852_{0.019}$ | $0.979_{0.003}$ | $0.874_{0.013}$ |
| Virchow-2 (Zimmermann et al., 2024) | $\mathbf{0.822}_{0.013}$ | $\underline{0.735}_{0.018}$ | $\mathbf{0.962}_{0.001}$ | $\mathbf{0.828}_{0.007}$ | $\underline{0.913}_{0.009}$ | $\underline{0.888}_{0.011}$ | $\mathbf{0.990}_{0.002}$ | $\underline{0.913}_{0.009}$ |
| Prov-GigaPath (Xu et al., 2024) | $0.784_{0.018}$ | $0.673_{0.022}$ | $0.941_{0.008}$ | $0.778_{0.019}$ | $0.885_{0.009}$ | $0.865_{0.016}$ | $0.981_{0.002}$ | $0.884_{0.009}$ |
| H-optimus-0 (Saillard et al., 2024) | $0.789_{0.031}$ | $0.650_{0.028}$ | $0.941_{0.012}$ | $0.779_{0.031}$ | $0.878_{0.010}$ | $0.838_{0.023}$ | $0.978_{0.003}$ | $0.876_{0.011}$ |
| **StainNet-Small (Ours)** | $0.709_{0.025}$ | $0.597_{0.036}$ | $0.892_{0.023}$ | $0.701_{0.025}$ | $0.874_{0.010}$ | $0.839_{0.020}$ | $0.979_{0.004}$ | $0.873_{0.010}$ |
| **StainNet-Base (Ours)** | $\underline{0.811}_{0.006}$ | $\mathbf{0.744}_{0.007}$ | $\underline{0.951}_{0.003}$ | $\underline{0.809}_{0.005}$ | $\mathbf{0.913}_{0.008}$ | $\mathbf{0.896}_{0.014}$ | $\underline{0.989}_{0.001}$ | $\mathbf{0.913}_{0.008}$ |

| Patch encoder | Linear probe | | | | MLP probe | | | |
|---|---|---|---|---|---|---|---|---|
| MIST | Acc. | Bal acc. | AUC | F1-score | Acc. | Bal acc. | AUC | F1-score |
| ResNet-50 (He et al., 2016) | $0.597_{0.009}$ | $0.592_{0.009}$ | $0.835_{0.002}$ | $0.593_{0.008}$ | $0.671_{0.005}$ | $0.667_{0.004}$ | $0.884_{0.002}$ | $0.671_{0.005}$ |
| CTransPath (Wang et al., 2022) | $0.663_{0.008}$ | $0.658_{0.008}$ | $0.878_{0.004}$ | $0.659_{0.006}$ | $0.732_{0.007}$ | $0.728_{0.007}$ | $0.920_{0.004}$ | $0.731_{0.007}$ |
| PathoDuet (Hua et al., 2024) | $0.590_{0.010}$ | $0.585_{0.011}$ | $0.827_{0.006}$ | $0.586_{0.013}$ | $0.664_{0.013}$ | $0.660_{0.012}$ | $0.885_{0.004}$ | $0.664_{0.011}$ |
| HIPT (Chen et al., 2022b) | $0.587_{0.007}$ | $0.581_{0.007}$ | $0.830_{0.006}$ | $0.577_{0.007}$ | $0.681_{0.006}$ | $0.676_{0.007}$ | $0.896_{0.004}$ | $0.676_{0.010}$ |
| Lunit (Kang et al., 2023) | $0.738_{0.006}$ | $0.733_{0.006}$ | $0.922_{0.003}$ | $0.736_{0.005}$ | $0.871_{0.005}$ | $0.869_{0.006}$ | $0.977_{0.001}$ | $0.871_{0.006}$ |
| UNI (Chen et al., 2024) | $0.773_{0.007}$ | $0.769_{0.007}$ | $0.940_{0.002}$ | $0.771_{0.008}$ | $0.874_{0.005}$ | $0.872_{0.005}$ | $0.979_{0.001}$ | $0.874_{0.005}$ |
| PathOrchestra (Yan et al., 2025) | $0.760_{0.008}$ | $0.755_{0.009}$ | $0.934_{0.003}$ | $0.758_{0.009}$ | $0.846_{0.004}$ | $0.843_{0.005}$ | $0.969_{0.001}$ | $0.845_{0.005}$ |
| GPFM (Ma et al., 2025) | $0.730_{0.009}$ | $0.725_{0.009}$ | $0.916_{0.003}$ | $0.726_{0.009}$ | $0.812_{0.006}$ | $0.808_{0.006}$ | $0.956_{0.002}$ | $0.811_{0.006}$ |
| UNI-2 (Chen et al., 2024) | $0.782_{0.006}$ | $0.778_{0.007}$ | $0.943_{0.002}$ | $0.782_{0.007}$ | $0.842_{0.002}$ | $0.838_{0.002}$ | $0.967_{0.001}$ | $0.841_{0.003}$ |
| Virchow-2 (Zimmermann et al., 2024) | $\underline{0.820}_{0.005}$ | $\underline{0.816}_{0.005}$ | $\underline{0.959}_{0.002}$ | $\underline{0.820}_{0.005}$ | $\underline{0.918}_{0.006}$ | $\underline{0.917}_{0.006}$ | $\underline{0.989}_{0.001}$ | $\underline{0.919}_{0.006}$ |
| Prov-GigaPath (Xu et al., 2024) | $0.803_{0.004}$ | $0.799_{0.004}$ | $0.951_{0.003}$ | $0.802_{0.004}$ | $0.873_{0.004}$ | $0.871_{0.005}$ | $0.978_{0.002}$ | $0.873_{0.005}$ |
| H-optimus-0 (Saillard et al., 2024) | $0.752_{0.007}$ | $0.748_{0.007}$ | $0.930_{0.001}$ | $0.753_{0.008}$ | $0.841_{0.003}$ | $0.838_{0.003}$ | $0.966_{0.002}$ | $0.841_{0.003}$ |
| **StainNet-Small (Ours)** | $0.779_{0.006}$ | $0.775_{0.006}$ | $0.941_{0.002}$ | $0.778_{0.007}$ | $0.884_{0.006}$ | $0.882_{0.006}$ | $0.982_{0.002}$ | $0.884_{0.006}$ |
| **StainNet-Base (Ours)** | $\mathbf{0.825}_{0.005}$ | $\mathbf{0.822}_{0.005}$ | $\mathbf{0.960}_{0.002}$ | $\mathbf{0.825}_{0.006}$ | $\mathbf{0.923}_{0.005}$ | $\mathbf{0.921}_{0.005}$ | $\mathbf{0.991}_{0.001}$ | $\mathbf{0.923}_{0.005}$ |

Table 3: Comparison results for our StainNet with 12 PFMs on the BCI and MIST classification datasets. We use linear probe and MLP probe methods for ROI-level fine-tuning and report four average evaluation metrics with standard deviation across all folds. The best result is in **bold** and the second best is underlined.

Table 2 shows the comparison results of ABMIL on *P53*-UCEC and *MLH1*-UCEC. We observe that the StainNet-Base achieves overall best performance on the three WSI tasks. We also report the paired t-test p-values between StainNet-Base and other PFMs on the *P53*-UCEC dataset, which are shown in Appendix C. For example, when compared with billion-scale models such as Prov-GigaPath and H-optimus-0, the 86M-parameter StainNet-Base still outperforms them on NTUH-Ki67-Liver by 0.83% and 3.38%. Moreover, it outperforms them by 10.9% and 15.2% in balanced accuracy on *MLH1*-UCEC. It is noteworthy that although StainNet-Small has extremely small parameters (only 22M), it remains highly competitive. For instance, on NTUH-Ki67-Liver, it achieves overall best performance compared to PFMs with similar parameter size, and outperforms UNI and GPFM (303M) by 0.13% and 1.38%. These results demonstrate the advantages of large-scale pre-training on non-H&E pathology images.

| Patch encoder | Glomerulus-Masson | | Glomerulus-PAS | | Glomerulus-PASM | |
|---|---|---|---|---|---|---|
| | *Bal acc.* | *F1-score* | *Bal acc.* | *F1-score* | *Bal acc.* | *F1-score* |
| ResNet-50 (He et al., 2016) | $0.253_{0.005}$ | $0.254_{0.024}$ | $0.336_{0.004}$ | $0.416_{0.004}$ | $0.258_{0.011}$ | $0.271_{0.013}$ |
| CTransPath (Wang et al., 2022) | $0.284_{0.022}$ | $0.292_{0.059}$ | $0.478_{0.010}$ | $0.551_{0.006}$ | $0.273_{0.027}$ | $0.273_{0.053}$ |
| PathoDuet (Hua et al., 2024) | $0.296_{0.028}$ | $0.315_{0.038}$ | $0.353_{0.016}$ | $0.431_{0.018}$ | $0.256_{0.019}$ | $0.277_{0.035}$ |
| HIPT (Chen et al., 2022b) | $0.319_{0.048}$ | $0.279_{0.062}$ | $0.528_{0.017}$ | $0.574_{0.012}$ | $0.296_{0.066}$ | $0.300_{0.076}$ |
| Lunit (Kang et al., 2023) | $0.380_{0.087}$ | $0.396_{0.163}$ | $0.607_{0.008}$ | $0.648_{0.009}$ | $0.361_{0.044}$ | $0.377_{0.132}$ |
| UNI (Chen et al., 2024) | $\underline{0.478}_{0.037}$ | $\underline{0.537}_{0.040}$ | $0.632_{0.030}$ | $\underline{0.665}_{0.023}$ | $0.390_{0.036}$ | $0.446_{0.033}$ |
| UNI-2 (Chen et al., 2024) | $0.395_{0.015}$ | $0.441_{0.015}$ | $0.535_{0.012}$ | $0.577_{0.012}$ | $0.301_{0.053}$ | $0.328_{0.074}$ |
| Virchow-2 (Zimmermann et al., 2024) | $0.359_{0.025}$ | $0.418_{0.030}$ | $0.555_{0.006}$ | $0.611_{0.006}$ | $0.403_{0.057}$ | $0.456_{0.067}$ |
| Prov-GigaPath (Xu et al., 2024) | $0.430_{0.058}$ | $0.484_{0.066}$ | $0.625_{0.019}$ | $0.661_{0.012}$ | $\mathbf{0.501}_{0.025}$ | $\mathbf{0.559}_{0.023}$ |
| H-optimus-0 (Saillard et al., 2024) | $0.427_{0.042}$ | $0.480_{0.052}$ | $\underline{0.633}_{0.014}$ | $0.660_{0.010}$ | $0.427_{0.042}$ | $0.480_{0.043}$ |
| **StainNet-Small (Ours)** | $0.351_{0.026}$ | $0.347_{0.099}$ | $0.592_{0.020}$ | $0.631_{0.014}$ | $0.335_{0.065}$ | $0.368_{0.106}$ |
| **StainNet-Base (Ours)** | $\mathbf{0.505}_{0.027}$ | $\mathbf{0.567}_{0.028}$ | $\mathbf{0.669}_{0.020}$ | $\mathbf{0.693}_{0.017}$ | $\underline{0.475}_{0.063}$ | $\underline{0.533}_{0.058}$ |

Table 4: Comparison of balanced accuracy and F1-score results for our StainNet with 10 PFMs on the Glomerulus-Masson, Glomerulus-PAS, and Glomerulus-PASM classification dataset. We use linear probe for ROI-level fine-tuning and report the average evaluation metrics with standard deviation across all random seeds. The best result is in **bold** and the second best is underlined.

## 4.3. Comparison results on ROI tasks

Table 3 and Table 4 present the comparative results on the IHC tasks and the special stain glomerulus tasks. For the IHC tasks, we observe that Virchow-2, which is pre-trained on both H&E and IHC images, and StainNet-Base both achieve superior overall performance compared to other models. However, considering that Virchow-2 is based on a ViT-Huge/14 architecture, its computational cost is substantially higher than that of StainNet-Base. Meanwhile, StainNet-Small also demonstrates a strong balance between computational efficiency and performance when compared with models of similar or even larger scales. It is worth noting that our goal is not to directly replace previous PFMs, but rather to highlight the advantages of pre-training on domain-specific pathology images. We further believe that training on larger-scale non-H&E datasets and using models with bigger parameter size will lead to further performance gains, which we leave as an important direction for future work. For the special stain tasks, we find that the advantage of StainNet-Small diminishes when compared with models of similar scale; however, StainNet-Base further widens the performance gap with other larger PFMs. We also provide Umap visualization for BCI in Appendix D.

## 4.4. Comparison of image retrieval

We also examine the retrieval capability of StainNet-Small and StainNet-Base to assess their semantic representation quality on special staining histology images. We compare StainNet with CTransPath and UNI on the MIST dataset, as shown in Figure 2. We observe that both StainNet models consistently achieve the best Recall@K and mAP@K

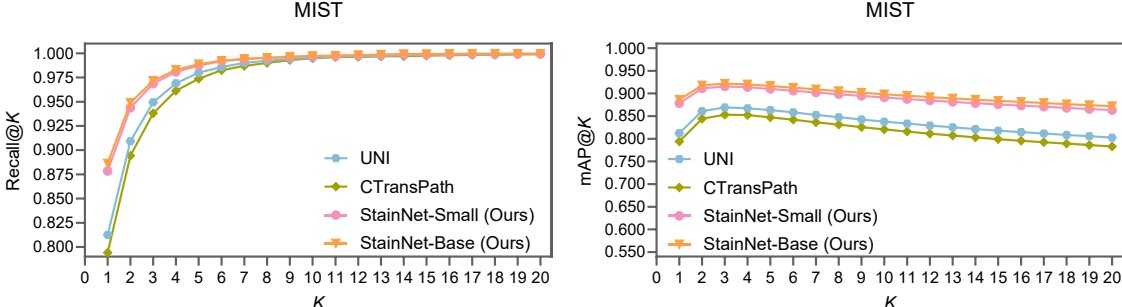

Figure 2: Comparison of image retrieval performance among CTransPath, UNI, and our StainNet-Small and StainNet-Base on the MIST ROI-level dataset.

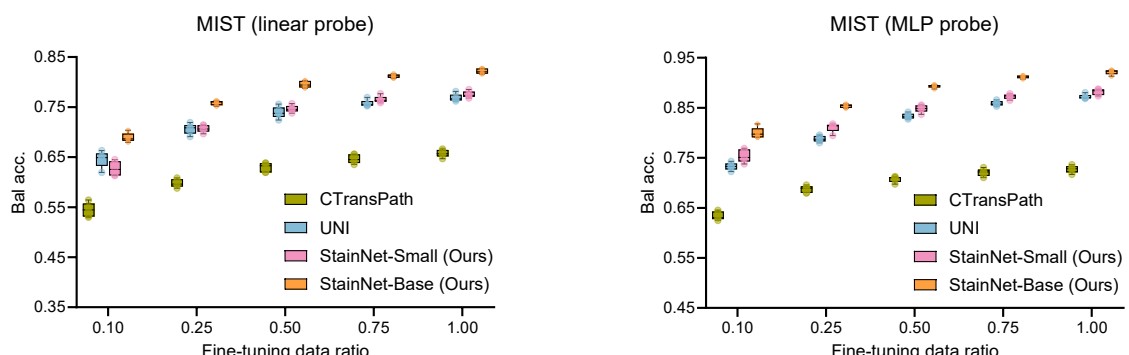

Figure 3: Linear probe and MIL probe fine-tuning results of CTransPath, UNI and our StainNet on the MIST ROI-level dataset across varying training data ratios.

across all choices of K, indicating that they have learned a well-structured embedding space for special staining images.

### 4.5. Effectiveness of different fine-tuning data ratios

We further evaluate the fine-tuning performance of StainNet models under varying amounts of downstream special-stain training data to assess their learning capability. Figure 3 presents the results of StainNet-Small, StainNet-Base, CTransPath, and UNI on the MIST dataset using linear and MLP probes under different proportions of fine-tuning samples. We observe that StainNet-Base achieves the best performance across all ratio settings. Although StainNet-Small performs slightly worse than UNI under the 10% data setting with a linear probe, it achieves the best performance in all other settings and substantially outperforms the CTransPath of similar parameter size. Notably, applying an MLP probe significantly increases the performance gap between StainNet and UNI in low-fine-tuning data scenarios.

| Datasets | CRC-100K | KatherMS | PANDA-*Karo.* |
|---|---|---|---|
| HIPT (Chen et al., 2022b) | $0.895_{0.001}$ | $0.554_{0.010}$ | $0.217_{0.005}$ |
| CTransPath (Wang et al., 2022) | $0.937_{0.001}$ | $0.688_{0.002}$ | $0.238_{0.009}$ |
| Lunit (Kang et al., 2023) | $0.910_{0.003}$ | $0.694_{0.003}$ | $0.270_{0.011}$ |
| UNI (Chen et al., 2024) | $0.937_{0.001}$ | $0.711_{0.004}$ | $0.535_{0.014}$ |
| **StainNet-Small (Ours)** | $0.931_{0.005}$ | $0.684_{0.001}$ | $0.252_{0.005}$ |
| **StainNet-Base (Ours)** | $0.930_{0.004}$ | $0.658_{0.006}$ | $0.322_{0.009}$ |

Table 5: Comparison of balanced accuracy for our StainNet with 4 PFMs on two H&E-stained ROI-level and one WSI-level classification dataset. We use linear probe for ROI-level and ABMIL for WSI-level fine-tuning, and report the average evaluation metrics with standard deviation across all random seeds. The best result is in **bold** and the second best is underlined.

### 4.6. Comparison results on H&E-stained tasks

We further explore the potential of the proposed StainNet models on H&E image tasks. Table 5 presents the quantitative results. We find that StainNet-Small is not significantly different from other PFMs with similar parameter sizes on both tasks, and that StainNet-Base does not improve performance on H&E image tasks despite having a larger parameter size than StainNet-Small. However, the large PFM UNI, specifically trained on H&E-stained images, leads in both task types, especially on the WSI task, where the gap is significant.

### 4.7. Comparison of different pre-training data ratios and iterations

We also conduct an ablation study to investigate the impact of different pre-training data scales. Specifically, we randomly sample 25 and 50 patches from each WSI to construct two pre-training datasets of different sizes, which are then used to pretrain StainNet-Small with DINO. Then we adapt the pre-trained models to the BCI and MIST ROI-level downstream tasks, as shown in Figure 4.a. We observe that larger pre-training datasets lead to stronger downstream performance, which validates the scalability of DINO on non-H&E images.

Finally, we investigate the performance variation of StainNet-Small under different pre-training iterations and the behavior of its loss curve. We believe these observations can provide valuable insights for the development of SSL methods in computational pathology, where pre-training details such as the expected loss curve and convergence behavior are often underreported. Figure 4.b and Figure 4.c illustrate the pre-training loss curve over 100 epochs and the corresponding linear probe performance on the MIST and Glomerulus-PAS dataset at six iteration checkpoints. We observe that on MIST, the loss plateaus between epoch 30 and 50, but with significant performance improvement during this period. After epoch 60, the overall loss begins to decrease again, but with slight changes in performance. Ultimately, the model reaches its optimal state at the final iteration. On Glomerulus-PAS, performance increases progressively with the number of epochs. These findings suggest that the shift in the DINO pre-training loss may serve as a valuable indicator of representation quality. While additional exploration is required to generalize these observations to other settings, we encourage future SSL development in computational pathology to con-

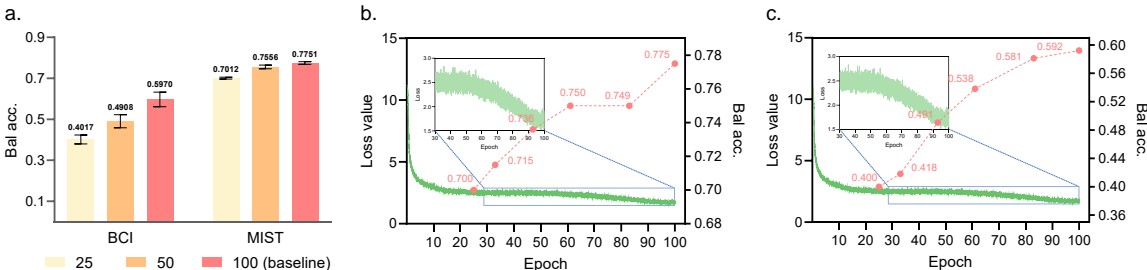

Figure 4: Pre-training loss curve of StainNet and comparison of linear probe performance on MIST under six different iterations.

sider allocating more training epochs, which may be crucial for achieving optimal model performance.

## 5. Conclusion and future works

In this work, we propose StainNet, a collection of pathology foundation models specifically for special staining histopathology image representation. Utilizing the DINO self-supervised framework, we train StainNet-Small and StainNet-Base models based on over 1.4 million unlabeled images from non-H&E WSIs in HISTAI. We evaluate StainNet models on three in-house IHC WSI tasks, two public IHC ROI tasks, and three in-house special stain ROI tasks, including classification, few-ratio settings, and image retrieval. The results demonstrate that StainNet models learn strong and robust representations for IHC and special stain pathology images.

For future work, we plan to develop larger StainNet models by expanding the pre-training dataset, scaling the model parameter size, and evaluating them on broader down-stream tasks. We will also explore integrating StainNet with H&E-based PFMs for multimodal applications such as survival prognosis and therapy response prediction. We believe that StainNet will further accelerate the research and applications in computational pathology involving diverse histological modalities.

## Acknowledgments

This work is supported by the National Natural Science Foundation of China (NSFC) (82430062), the Shenzhen Engineering Research Centre (XMHT20230115004), and the Tsinghua Shenzhen International Graduate School Cross-disciplinary Research and Innovation Fund Research Plan (JC2024002). We thank Qiehe Sun for providing GPU support.

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

# Appendix A. Details of pre-training datasets

We observe that the non-H&E WSI data collected in HISTAI exhibit a certain degree of ambiguity in naming conventions. Specifically, heterogeneous clone identifiers and inconsistent file naming rules are used across different cases, and some WSIs lack explicit annotations of their corresponding non-H&E staining types. As a result, it is practically challenging to assign fully accurate and unified stain labels to all WSIs. Nevertheless, we made systematic efforts to curate and organize the data to the best extent possible, with the goal of providing the community with a reasonably accurate distribution for the pre-training data, thereby reflecting the overall composition and diversity of non-H&E staining modalities. Based on our statistics, the HISTAI non-H&E pre-training dataset comprises 331 distinct immunohistochemistry (IHC) staining types and 12 special staining types. Figure 5 illustrates the overall distribution of different staining modalities within the pre-training data. Furthermore, Table 6 reports the distribution of IHC staining types with more than 20 cases, highlighting the major IHC sources contributing to the pre-training process. Table 7 summarizes the case distribution of all special staining types. We hope that these statistics and visualizations will serve as a useful reference for future studies leveraging HISTAI non-H&E data.

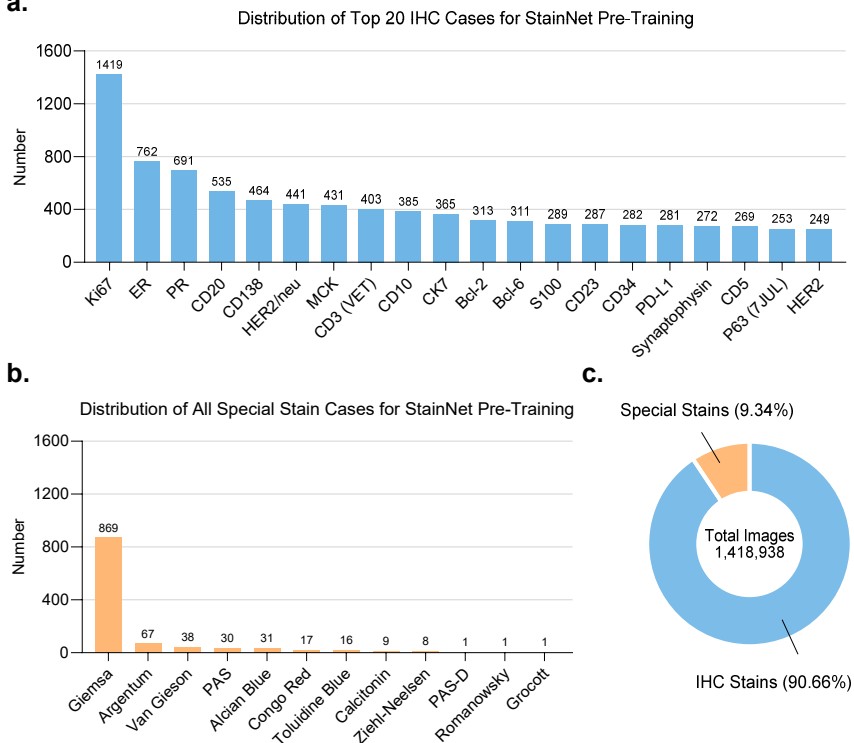

Figure 5: Overview of the StainNet pre-training dataset. **a.** Distribution of the top 20 IHC stains by case count. **b.** Distribution of the all special stains by case count. **c.** Distribution of patch image types used for training.

| IHC stain type | Case number | Patch image number |
| --- | --- | --- |
| Ki67 | 1419 | 95715 |
| ER | 762 | 51533 |
| PR | 691 | 46757 |
| CD20 | 535 | 39513 |
| CD138 | 464 | 41006 |
| HER2/neu | 441 | 22850 |
| MCK | 431 | 28564 |
| CD3 (VET) | 403 | 29128 |
| CD10 | 385 | 35373 |
| CK7 | 365 | 25864 |
| Bcl-2 | 313 | 28075 |
| Bcl-6 | 311 | 27345 |
| S100 (4c4) | 289 | 24579 |
| CD23 | 287 | 24359 |
| CD34 | 282 | 23313 |
| PD-L1 | 281 | 25500 |
| Synaptophysin | 272 | 18537 |
| CD5 | 269 | 20015 |
| P63-7JUL | 253 | 14431 |
| HER2 | 249 | 8607 |
| Cyclin-D1 | 247 | 22386 |
| MUM1-Protein | 233 | 20339 |
| TTF-1 | 230 | 13932 |
| CK20 (ks20) | 214 | 14266 |
| CD30 | 208 | 17529 |
| SMA | 202 | 18070 |
| PAX-8 | 190 | 14440 |
| Chromogranin-A | 185 | 13375 |
| CD45 | 183 | 16063 |
| GATA-3 | 178 | 11796 |
| CDX2 | 174 | 10253 |
| Desmin | 154 | 10415 |
| MSH6 | 154 | 10044 |
| MlH1 | 151 | 11546 |
| SOX-10 | 142 | 14610 |
| MSH2 | 141 | 11224 |
| PAX-5 | 138 | 10074 |
| PMS2 | 136 | 10648 |
| CD56 | 128 | 9803 |
| Vimentin | 126 | 11039 |
| Melan-A | 124 | 10128 |

| IHC stain type | Case number | Patch image number |
|---|---|---|
| P53 | 123 | 9530 |
| P16 | 119 | 7881 |
| HMB45 | 116 | 9334 |
| CD56 (123c3) | 112 | 7871 |
| P40 | 112 | 6112 |
| AMACR | 112 | 4823 |
| Epstein-Barr-Birus | 111 | 10141 |
| CK14 | 104 | 5918 |
| CD117 | 100 | 8255 |
| WT1-WT49 | 100 | 7267 |
| CD68 | 99 | 8079 |
| E-Cadherin | 92 | 4808 |
| CK5&6 | 90 | 5444 |
| S100 | 81 | 7301 |
| CD15 | 79 | 6646 |
| CD4 | 70 | 5575 |
| Kappa-Light-Chain | 67 | 5637 |
| Lambda-Light-Chain | 65 | 4974 |
| EMA (gp1) | 63 | 5318 |
| CD8 | 61 | 4992 |
| A1K | 59 | 4542 |
| DOG-1 | 58 | 4482 |
| CD79A | 56 | 4088 |
| CD99 | 54 | 5074 |
| Calponin-1 | 54 | 4412 |
| Calretinin | 52 | 4395 |
| Arginase-1 | 49 | 4299 |
| SATB2 | 49 | 3918 |
| CD16 | 47 | 4400 |
| ERG | 47 | 3070 |
| HlA-DR-Antigen | 46 | 3525 |
| Glypican-3 | 46 | 3104 |
| CK8 & CK18 (b22) | 46 | 2957 |
| Napsin-a | 46 | 2906 |
| INSML | 43 | 3268 |
| CK-HMW | 43 | 1532 |
| GFAP | 42 | 3566 |
| P63 | 42 | 3167 |
| CD31 | 40 | 3207 |
| CD2 | 38 | 3012 |
| Beta-Catenin | 37 | 3231 |
| ER-other | 37 | 3037 |
| CD21 | 36 | 2869 |

| IHC stain type | Case number | Patch image number |
|---|---|---|
| INI-1 | 35 | 3383 |
| CD7 | 35 | 2468 |
| TDT | 32 | 2934 |
| CK19 | 32 | 2278 |
| Androgen-Receptor | 32 | 1747 |
| Inhibin-Alpha | 29 | 2646 |
| Mammaglobin | 29 | 2089 |
| PSA | 29 | 1796 |
| C-MYC | 27 | 2393 |
| HHV8 | 27 | 2067 |
| CD3-K | 27 | 1636 |
| Myeloperoxidase | 26 | 2424 |
| CD61 | 26 | 2096 |
| CK20 (KS20) | 26 | 2071 |
| STAT6 | 26 | 1992 |
| MDM2 | 26 | 1850 |
| CD38 | 26 | 1643 |
| Heppar | 25 | 2213 |
| PLAP | 25 | 1968 |
| LIFR | 23 | 2249 |
| IGG4 | 23 | 2106 |
| Caix-TH22 | 23 | 2059 |
| CK-5-6-d5-16b4 | 23 | 1653 |
| CEA | 22 | 1677 |
| Alpha-Fetoprotein | 21 | 1850 |
| SAll4 | 21 | 1504 |
| CD3 | 21 | 1453 |
| Podoplanin | 20 | 1845 |
| Total | 16150 | 1216942 |

Table 6: IHC stain type distribution of our data for StainNet pre-training.

| Special stain type | Case number | Patch image number |
|---|---|---|
| Giemsa | 869 | 114016 |
| Argentum | 67 | 2956 |
| Van Gieson | 38 | 5039 |
| PAS | 30 | 3061 |
| Alcian Blue | 31 | 3743 |
| Congo Red | 17 | 763 |
| Toluidine Blue | 16 | 1011 |
| Calcitonin | 9 | 816 |
| Ziehl Neelsen | 8 | 881 |
| PAS-D | 1 | 100 |
| Romanovsky | 1 | 100 |
| Grocott | 1 | 14 |
| Total | 1088 | 132500 |

Table 7: Special stain type distribution of our data for StainNet pre-training.

## Appendix B. Details of DINO pre-training settings

We provide the data augmentation strategy and detailed hyperparameter settings for DINO, as shown in Table 8 and Table 9.

| Augmentation | Value |
|---|---|
| Image size | $224 \times 224$ |
| Local view size | $96 \times 96$ |
| Global random resize scale | (0.4, 1.0) |
| Local random resize scale | (0.05, 0.4) |
| Number of local views | 8 |
| Random horizontal flip | 0.5 |
| Random vertical flip | 0.0 |
| Random gray scale | 0.2 |
| Color jitter brightness | 0.8 |
| Color jitter contrast | 0.8 |
| Color jitter hue | 0.2 |
| Color jitter prob | 0.8 |
| Color jitter saturation | 0.4 |
| Color jitter strength | 0.5 |
| Gaussian blur limit | 0 |
| Gaussian blur prob | 1.0 |
| Gaussian blur sigmas | (0.1, 2.0) |
| Global view Gaussian blur prob | 0.1 |
| Global view Solarize prob | 0.2 |
| Global view Solarize threshold | 0.5 |
| Local view: Gaussian blur prob | 0.5 |
| Normalization mean | (0.5, 0.5, 0.5) |
| Normalization std | (0.5, 0.5, 0.5) |

Table 8: DINO data augmentation settings used in StainNet pre-training.

We provided some examples of original images and their augmented versions, which are illustrated in Figure 6.

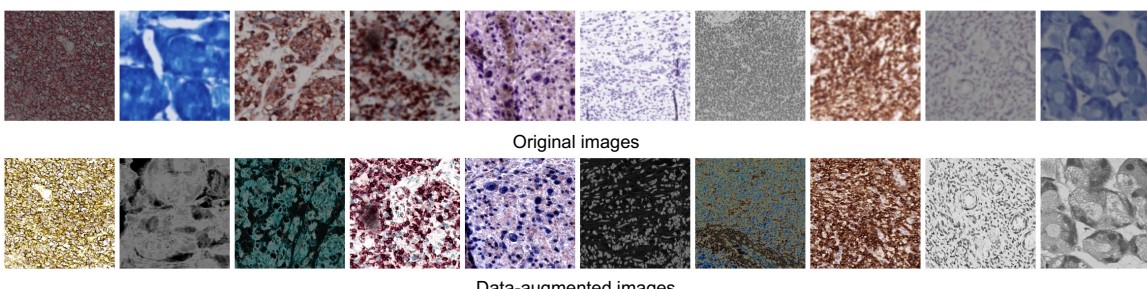

Original images

Data-augmented images

Figure 6: A set of original special staining images and their data-augmented versions.

| Hyperparameter | Value |
|---|---|
| Batch normalization | False |
| Bottleneck dimension | 256 |
| Center momentum | 0.9 |
| Hidden dimension | 2048 |
| Learning rate scale method | linear |
| Momentum end | 1.0 |
| Momentum start | 0.9995 |
| Last layer normalization | True |
| Output dimension | 65536 |
| Epoch | 100 |
| Total batch size | 256 |
| Student freeze last layer epochs | 0 |
| Student temperature | 0.1 |
| Teacher temperature | 0.04 |
| Warmup teacher temperature | 0.04 |
| Warmup teacher temperature epochs | 0 |
| Weight decay end | 0.4 |
| Weight decay start | 0.04 |
| Optimizer | AdamW |
| Optimizer betas | (0.9, 0.999) |
| Optimizer eps | 1e-8 |
| Optimizer learning rate | 5e-4 |
| Optimizer weight decay | 0.04 |
| Automaticmixedprecision | BF16 |

Table 9: DINO hyperparamters used in StainNet pre-training. StainNet-Samll is trained using $1 \times$ 96GB NVIDIA PRO 9000, and StainNet-Base is trained using $8 \times$ 80GB NVIDIA A100.

# Appendix C.  Statistical significance analysis

| Model | Acc. $(p)$ | Bal acc. $(p)$ | AUC $(p)$ | F1-score $(p)$ |
|---|---|---|---|---|
| ResNet-50 | 2.70e-04*** | 3.34e-03** | 1.16e-03** | 1.03e-04*** |
| CTransPath | 2.40e-01 | 3.13e-02* | 1.36e-01 | 3.85e-02* |
| PathoDuet | 8.26e-03** | 2.57e-02* | 2.37e-03** | 5.93e-03** |
| HIPT | 4.66e-02* | 2.48e-02* | 3.08e-02* | 4.43e-02* |
| Lunit | 3.27e-02* | 2.85e-02* | 8.04e-04*** | 7.86e-03** |
| UNI | 9.94e-01 | 6.03e-02 | 4.93e-01 | 1.40e-01 |
| PathOrchestra | 3.68e-02* | 5.65e-02 | 2.23e-02* | 3.61e-02* |
| GPFM | 9.94e-01 | 6.03e-02 | 1.77e-01 | 1.44e-01 |
| UNI-2 | 8.96e-02 | 3.40e-02* | 7.33e-03** | 6.78e-03** |
| Virchow-2 | 3.33e-02* | 3.89e-02* | 6.68e-02 | 3.13e-03** |
| Prov-GigaPath | 1.61e-02* | 1.12e-01 | 4.99e-01 | 6.55e-02 |
| H-optimus-0 | 1.78e-01 | 4.66e-02* | 9.12e-02 | 4.50e-02* |
| StainNet-Small | 9.83e-02 | 2.60e-02* | 2.10e-03** | 1.72e-02* |

Table 10: Paired t-test p-values between StainNet-Base and different PFMs on the *P53-UCEC* dataset (* $p < 0.05$, ** $p < 0.01$, *** $p < 0.001$).

## Appendix D. Feature visualization

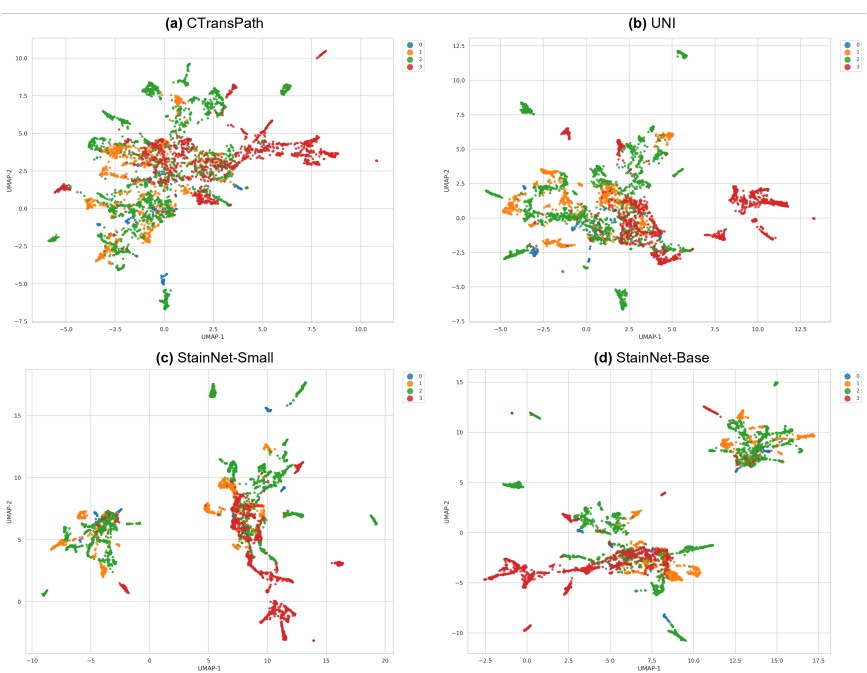

Figure 7: Umap visualization of high-dimension feature

