# OpenReview forum: "StainNet: Scaling Self-Supervised Foundation Models on Immunohistochemistry and Special Stains for Computational Pathology"
_MIDL.io/2026/Conference — MIDL 2026 Poster_

### Official Review · Reviewer_FyLQ · 2026-01-06

**Confidence:** 5
**Preliminary Rating:** 4
**Final Rating:** 5

**Summary:**

The authors present a specialized foundation model StainNet, designed for histopathology images stained with special stains such as IHC, Masson’s trichrome etc. Utilizing the DINO self-distillation framework, the authors pre-trained StainNet on a large-scale dataset of ~1.4 million patches extracted from 20,231 WSIs from the HISTAI database.
The authors demonstrate significant performance improvements across multiple tasks including slide-level Ki67 liver malignancy classification and ROI-level tasks such as HER2 expression and biomarker identification, outperforming existing foundation models by substantial margins. This work is particularly significant as it establishes a new benchmark for special staining image analysis in computational pathology, addressing an important gap where H&E-based models often fail due to fundamental differences in staining mechanisms and visual appearance.

**Strengths:**

1. The paper has a compelling novelty. While many existing PFMs focus exclusively on H&E-stained images, StainNet targets special stain images like IHC which are clinically essential but often overlooked in foundation model development.
2. The authors provide a comprehensive evaluation across multiple tasks (slide-level classification, ROI-level analysis, and image retrieval) with rigorous comparisons against both similar-sized and larger PFMs. The paper's evaluation methodology is robust, using stratified 5-fold cross-validation across multiple metrics (balanced accuracy, AUC, F1-score) and comparing against various patch encoders. They demonstrate that representations learned from special stain images provide substantial advantages over H&E-based PFMs when applied to special-stain tasks, with StainNet achieving 87.69% average balanced accuracy on the NTUH-Ki67-Liver task compared to 79.80% for ResNet-50 and 84.7% for Lunit.
3. Their methodology is well-structured, leveraging an already established DINO framework effectively for self-supervised learning on special staining images.
4. The paper also provides valuable implementation details including hyperparameters, training procedures, and evaluation metrics that will be useful for reproducibility.
5. The model is built with a robust, yet lightweight 22 million parameter backbone, making it also ideal for deployment in resource-constrained environments.
6. Additionally, the authors provide public access to their model weights on Hugging Face, which will facilitate further research in this domain.

**Weaknesses:**

1. While the pre-training dataset is large, the term "special stains" can cover a highly diverse set of stains. The paper do not provide a detailed breakdown of the specific proportions of different stain types (e.g., Masson’s trichrome, various IHC markers) that are more prevalent in HistAI dataset. Without this, it’s hard to tell if the model is truly "generalist" for all special stains or if its performance is mostly driven by a few dominant stain types.
2. The paper compare StainNet against several H&E-centric PFMs such as PathOrchestra, UNI, etc. However, the work fails to reference and evaluate against more recent state-of-the-art models, such as Prov-GigaPath, UNI2-h or Virchow2, which are known to incorporate both H&E and IHC data during pre-training.

**Detailed Comments:**

1. While the paper mentions following the DINO approach for global and local crops, clarifying the exact color jittering, solarization, or staining-specific augmentations used would be valuable for the sake of reproducibility.

2. The authors specified using the DINO self-supervised framework, but it is not clear whether they are using DINOv1 or DINOv2. This clarification would be helpful.

3. In Figure 6, authors note the loss plateaus while performance improves for MIST dataset. Was this observed across other tasks as well?

4. In Section 4.4, the authors mention that the high retrieval scores indicate a "well-structured embedding space". To make this claim stronger, it would be great to see t-SNE or UMAP visualizations. Comparing how StainNet clusters special stain features versus an H&E-based model would provide a very clear visual proof of the model's superior representation capability.

**Justification Of Final Rating:**

The authors have thoroughly addressed all concerns through substantial revisions. Specifically, the inclusion of Base model variant with their Small model, alongside a granular breakdown of the pre-training dataset distributions, provides the technical depth and transparency that were previously missing. The expanded evaluation incorporating more recent PFMs, additional IHC and special-stain downstream tasks, and statistical significance analysis, robustly demonstrates the model’s stability and generalizability.
Furthermore, the addition of UMAP visualizations on the BCI dataset provides valuable qualitative evidence of the model's representation capabilities. These improvements, combined with the comprehensive reproducibility details, significantly elevate the paper’s impact. While minor clarifications regarding feature extraction (e.g., CLS vs. mean pooling) and the impact of amount of training data would be beneficial, the overall novelty in specialized application to IHC and special stains are compelling. Therefore, I am increasing my recommendation to a Strong Accept.

**Justification Of The Preliminary Rating:**

The StainNet paper provides an impactful contribution by specifically targeting special-stain images, a critical yet often underserved domain in foundation model development. The empirical results are convincing, achieving an 87.69% balanced accuracy on specialized tasks and also the release of model weights provides immediate utility to the computational pathology community.
However, the preliminary rating is Weak Accept due to few primary technical gaps that prevent Strong Accept.
First, since the term "special stains" is quite broad; the lack of a granular breakdown of proportions of the ~20,000 WSIs in the pre-training dataset makes it difficult to assess the model's true generalizability across diverse stains like Masson’s trichrome vs various IHC markers. Second, the evaluation lacks additional critical benchmarks against the most recent state-of-the-art PFMs which are known to incorporate non-H&E data. Addressing these concerns would improve the paper's standing significantly.

**Questions To Address In The Rebuttal:**

1. Can authors please provide a breakdown of the WSIs belonging to specific staining types or IHC markers that constitute the pre-training dataset.

2. Please consider comparing StainNet against newer-generation models like Prov-GigaPath, UNI2 or Virchow2, which also incorporate IHC data.

---

> ### Author Response · Authors · 2026-01-25
> **Response for Reviewer FyLQ**
>
> We appreciate the valuable comments from Reviewer FyLQ. We have made comprehensive revisions to the manuscript to address all concerns to the best of our ability. All changes in the revised manuscript are highlighted in **green**. Our responses are provided below.
>
> **W1:** Thank you for your valuable comment. We have added further details regarding the HISTAI pre-training data in the revised manuscript. We acknowledge that the existing pre-training data still suffers from insufficient diversity and uneven distribution of data types. In the revised manuscript, we added two WSI-level IHC datasets and three special-stain glomerular classification datasets to further explore the potential of the proposed StainNet and the existing PFM in non-H&E images. In future work, we will further enrich the special-stain pre-training data and explore more diverse downstream tasks.
>
> **W2:** We appreciate the reviewers' valuable concern. We have added comparisons of Prov-GigaPath, UNI-2, Virchow-2, and H-Optimus-0 to the revised manuscript. We have also trained an additional ViT-Base framework, StainNet-Base, to further demonstrate the model's advantages under self-supervised training on special staining and IHC data.
>
> **Comment 1:** Thank you for the reviewers' questions. We have added all the data augmentation techniques used in the DINO method to the revised appendix manuscript.
>
> **Comment 2:** Thank you for the reviewer's questions. Due to limited computing resources, we used the self-supervised learning method DINOv1 for efficient training and evaluation. We will further explore the potential of DINOv2 in non-H&E images in our future work.
>
> **Comment 3:** We have added performance improvements in a glomerular task to the revised manuscript.
>
> **Comment 4:** We have added feature visualizations of StainNet-Small, StainNet-Base, CTransPath, and UNI in MIST's Umap.
>
> **Q1:** As shown in W1, we added the details of the pre-training data to the revised manuscript.
>
> **Q2:** As shown in W2, we have added comparison results with newer-generation models.

---

### Official Review · Reviewer_1oKo · 2026-01-10

**Confidence:** 5
**Preliminary Rating:** 3

**Summary:**

This paper proposes StainNet, a ViT-Small (≈22M params) pathology foundation model pretrained specifically on non-H&E (“special stain”) WSIs using DINO self-distillation self-supervised learning. The authors curate 20,231 special-stain WSIs from the public HISTAI database and sample ~1.42M 224×224 patches for SSL pretraining (Sections 3.1–3.2, Fig. 1). Downstream evaluation is performed on (i) an in-house Ki67 liver WSI 3-class classification task using multiple MIL heads, and (ii) two public ROI-level IHC datasets (BCI: HER2 scoring; MIST: biomarker classification) using linear/MLP probes (Section 4, Tables 1–2). The results indicate StainNet consistently outperforms several popular H&E-centric PFMs of similar size (CTransPath, HIPT, Lunit, PathoDuet) and, in the reported settings, can outperform some larger general PFMs (Fig. 3). The paper additionally reports retrieval results on MIST (Fig. 4), data-ratio experiments (Fig. 5), and an analysis relating pretraining loss dynamics to downstream performance (Fig. 6).

**Strengths:**

1. Clear and clinically motivated problem: The limitation of H&E-pretrained PFMs for special stains (notably IHC) is well-motivated and relevant to real workflows.
2. Simple, reproducible training recipe: Using a standard, well-understood SSL method (DINO) on a reasonably large dataset makes the approach easy to adopt/extend (Section 3.2).
3. Strong empirical gains on reported tasks: Across multiple MIL heads on the Ki67 liver WSI task (Table 1) and across both probe settings on BCI/MIST (Table 2), StainNet is consistently best or tied-best.
4. Breadth of evaluation “modes”: The paper goes beyond classification to include retrieval, data-ratio robustness, and pretraining-iteration analysis (Figs. 4–6), which collectively strengthen the representation-learning narrative.
5. Comparison against both similar-scale and larger PFMs: The explicit comparison to larger models (e.g., UNI / others shown in Fig. 3) is helpful and reinforces the domain-specific pretraining argument.
6. Model release: Public release of weights is valuable for the community and improves practical impact.

**Weaknesses:**

1. Limited methodological novelty: The core approach is essentially “ViT-S + DINO on special-stain patches.” The main contribution is the domain-specific pretraining and empirical validation rather than new SSL methodology.

2. Downstream evaluation is narrow relative to the claim “special stains”: Despite positioning as a foundation model for special staining broadly, the downstream tasks are overwhelmingly IHC-centric (Ki67, HER2, ER/PR). There is little evidence for stains like Masson’s trichrome, PAS, GMS, etc., even though these are explicitly discussed as motivation.

3. Potential confounds in ROI tasks (especially MIST): Treating “biomarker type” (HER2 vs Ki67 vs ER vs PR) as a 4-class classification task may be partially solvable via stain protocol/style cues rather than morphology/semantic understanding. This risks overstating “representation quality” unless additional controls are included.

4. Insufficient details to assess leakage and split rigor:

* For the in-house WSI dataset: no patient-level split discussion (could multiple slides per patient exist?).

* For BCI/MIST: unclear if ROIs from the same WSI/patient appear in multiple folds.

* Early stopping/model selection procedure is underspecified; it’s unclear what constitutes validation vs test within each CV fold (Section 4.1).

5. Dataset description inconsistencies / missing metadata:

* BCI is stated as 4,870 images, but the class counts listed sum to 4,890 (240+1,153+2,142+1,355).

* HISTAI “special stains” are treated as a single pool; there is no breakdown by stain type, organ, scanner, resolution/magnification—critical for understanding what the model actually learned and how it may generalize.

6. Fairness/standardization across PFMs not fully established: For large PFMs in particular, differences in recommended preprocessing, magnification assumptions, tokenization/patchification, and feature extraction pipelines can affect outcomes. The paper states “all baseline methods are trained using the same settings” (Section 4.1), but more detail is needed to ensure comparisons are apples-to-apples.

**Detailed Comments:**

* Tighten the claim scope or expand evidence. If StainNet is marketed as special stain foundation model, please either:
  * add at least one non-IHC downstream benchmark (e.g., fibrosis staging / collagen quantification on Masson’s trichrome; PAS for kidney structures; GMS; etc.), or
  * explicitly scope the claim to IHC-focused foundation modeling, and reflect this in the title/abstract.

* Clarify what “special stain” means operationally in HISTAI. Provide a table: stain categories included, counts of WSIs per stain, and whether any “near-H&E” variants are included.

* Patch sampling strategy needs justification and ablation. The method samples up to 100 random non-overlapping 224×224 patches per WSI. This equalizes slides but may underrepresent heterogeneity in large slides and overrepresent small tissue fragments. Suggested additions:
  * A small ablation: 25/50/100/200 patches per WSI (or fixed total patches) and impact on downstream.
  * Describe whether you filter out background after Otsu, and whether patch-level tissue percentage thresholds are applied.

* Report magnification / mpp normalization. 224×224 pixels can correspond to very different physical sizes across scanners. Please report:
extraction level (20×/40×?), or microns-per-pixel used, and whether WSIs are rescaled to a common resolution before patching.

* Define the cross-validation protocol precisely. For each dataset, specify:

  * what unit is split (patient / slide / ROI),

  * whether multiple ROIs from one slide can occur in different folds (should be avoided),

  * how early stopping is done (validation split within training folds?) and what metric selects the checkpoint. This is essential for trusting the reported improvements.

* Define AUC computation for multiclass tasks. Is AUC macro-averaged one-vs-rest? weighted? per-class then averaged? This affects interpretability (Table 2).

* Add statistical testing or confidence intervals for key comparisons. reporting mean±std across folds is good; consider paired tests across folds for the most important head-to-head comparisons (e.g., StainNet vs Lunit, and StainNet vs UNI).

* Strengthen the “domain vs scale” conclusion with a stronger control. A very informative experiment would be:

  * “continued SSL” (domain-adaptation) where a strong H&E PFM is continued-pretrained on the same special-stain patch set with the same DINO objective (or a comparable SSL objective).

* Ensure preprocessing parity across models. Provide explicit preprocessing for each PFM:

  * stain normalization/no normalization,

  * image normalization stats,

  * how features are extracted (CLS token? mean pooled? last layer?).

* Control for stain-style shortcuts. For the MIST “biomarker type” classification:

  * Add a baseline that uses color histograms / low-level features or a shallow CNN to quantify how much of the task is solvable by stain appearance alone.

  * Alternatively, consider a more morphology-driven task (e.g., Ki67 positivity estimation, or cell-level classification) if labels exist.

* Include qualitative retrieval examples. Since retrieval is a key claim (Fig. 4), show a few query–topK retrieval grids comparing StainNet vs a baseline to verify the model retrieves semantically similar tissue regions rather than just color/style matches.

* Fix dataset count inconsistency (BCI) and clarify whether images are real vs generated/augmented (BCI is introduced from an image generation paper, which raises questions about the nature of the data used for classification).

* Minor: Figure 5 caption appears to say “MIL probe” in at least one place, but it likely means MLP probe—please standardize terminology.

**Justification Of The Preliminary Rating:**

The paper tackles an important gap—foundation models for special-stain pathology—and shows consistent empirical gains over H&E-pretrained PFMs. However, the contribution is largely empirical with limited methodological novelty, and key details on data splits, confound control, and generality beyond IHC are insufficiently specified, reducing confidence in the claims.

**Questions To Address In The Rebuttal:**

* What is the breakdown of the 20,231 WSIs by stain type (IHC vs trichrome vs PAS vs others), organ site, and scanner/resolution?

* At what magnification (or mpp) were patches extracted? Were WSIs rescaled to a common mpp before sampling?

* Data leakage control: Are CV splits at patient level for NTUH-Ki67-Liver? For BCI and MIST, do ROIs from the same WSI/patient ever appear across folds?

* During 5-fold CV, how is early stopping done without peeking at the test fold? Is there a separate validation split inside each training fold?

* Why do the reported class counts sum to 4,890 while the dataset is described as 4,870? Which is correct?

* How is multiclass AUC computed and averaged in Tables 1–2?

* What exact feature extraction pipeline (layer, pooling, normalization) was used for UNI / PathOrchestra / GPFM comparisons?

* Why more rececent pathology foundation models such as Gigapath and H-Optimus-0 are not included in the benchmarks? Also, UNI2 performance comparison should be included.

* Potential confounding in MIST biomarker classification: Have you tested a simple color/texture baseline to estimate whether the task can be solved by stain appearance alone?

* How sensitive are results to (i) number of patches per WSI during pretraining, and (ii) ImageNet initialization vs training from scratch?

---

> ### Author Response · Authors · 2026-01-25
> **Response for Reviewer 1oKo**
>
> We appreciate the valuable comments from Reviewer 1oKo. We have made comprehensive revisions to the manuscript to address all concerns to the best of our ability. All changes in the revised manuscript are highlighted in **blue**. Our responses are provided below.
>
> **W1:** We appreciate the reviewer's perspective on the methodological novelty. Our initial aim was to explore the potential of self-supervised learning on non-H\&E images, and DINO is a convenient and effective approach for this. In the revised manuscript, we further use DINO to train a ViT-Base foundation model called StainNet-Base to further highlight the value of the entire work.
>
> **W2:** Thank you for the valuable comments. To further evaluate the representation performance of StainNet on the special stain images, we have added three ROI-level Masson, PAS, and PASM stained glomerular classification tasks. Detailed introductions to the datasets and their results are shown in the revised manuscript.
>
> **W3:** Thank you for your concern about the ROI task. We have added additional downstream datasets to support our work.
>
> **W4:** We thank the reviewer for the concerns about the implementation details. For the dataset splitting, each WSI in our in-house datasets comes from an independent patient case. For the public BCI and MIST dataset, their official descriptions do not mention WSI-level or patient-level splitting methods, therefore it may be possible that ROI data in different folds originate from the same WSI or patient. For NTUH-Ki67-Liver, BCI, and MIST, we use a 5-fold train-val cross-validation approach. This involved dividing the datasets into 5 groups based on label stratification, and for each fold, using four groups as the training set and one group as the validation set for experiments. We use balanced accuracy as the evaluation metric for model selection, with an early stop of 5 epochs. If the highest balanced accuracy is not achieved for 5 consecutive epochs during training, model training is stopped, and the fold ends and reports the best balanced accuracy as well as corresponding accuracy, AUC, and F1-score. We have provided further detailed explanations in the revised manuscript to support our work.
>
> **W5:** We thank you for your careful review. The number of images in the BCI dataset is a typo; it actually contains 4870 images, including 240 in category 0, 1153 in category 1+, and 1335 in category 2+. For the HISTAI pre-trained dataset, we have added more detailed illustrations and related descriptions in the revised manuscript to show its staining type and other information.
>
> **W6:** We appreciate your valuable comments. For all PFMs, we used the official preprocessing methods for each model. For the baseline setting, we provide further details here: For the WSI-level dataset, we first performed foreground segmentation on each WSI using OTSU, then cropped 224*224 patches at 20x magnification. These patches were then individually converted into high-dimensional feature vectors using each PFM's independent image-processing method.

---

> > ### Author Response · Authors · 2026-01-25
> >
> > **Comment 1:** We have added three additional non-IHC downstream benchmarks and changed the title to "StainNet: Scaling Self-Supervised Foundation Models on Immunohistochemistry and Special Stains and for Computational Pathology" to better reflect our work. We have also improved some parts of the abstract.
> >
> > **Comment 2:** We have added tables and figures to the appendix of the revised manuscript to provide details on the pre-training data.
> >
> > **Comment 3:** In our revised manuscript, we added ablation experiments targeting pre-trained data sizes (25% and 50%) to evaluate the data-scaling advantage. We used OTSU to first filter the background before performing random sampling. We used a tissue percentage threshold of 0.3, meaning that if the tissue percentage exceeded 30% of the image, it would be used for random sampling.
> >
> > **Comment 4:** We used 20x magnification for patch cutting, and we did not change any physical dimensions of the WSI before patching.
> >
> > **Comment 5:** We have described the experimental setting and dataset splitting in further detail in the revised manuscript.
> >
> > **Comment 6:** We adopt macro-averaged one-vs-rest AUC, implemented using the roc_auc_score function in scikit-learn library (average='macro', multi_class='ovr').
> >
> > **Comment 7:** We have added t-test experiments to the revised manuscript.
> >
> > **Comment 8:** Due to time constraints, we were unable to validate the continuous SSL on the existing PFM. However, we trained a StainNet-Base, a larger-scale model, to demonstrate its ability.
> >
> > **Comment 9:** We did not use coloring normalization techniques to preprocess the images. Furthermore, for each PFM, we used its own official preprocessing mode, so the normalized transform values may differ across PFMs. All ViT-based PFMs used CLS tokens for feature extraction, while ResNet and CTransPath (SwinTransformer) PFMs used global mean pooling.
> >
> > **Comment 10:** We understand your concerns about evaluating the model only on the MIST task. We have added more downstream experiments to the revised manuscript to validate the model's effectiveness further.
> >
> > **Comment 11:** Due to time constraints, we can only present the additional quantitative retrieval results on StainNet-Base for now, and we also include the UMAP feature visualization for the MIST task in the revised manuscript.
> >
> > **Comment 12:** We have corrected the issue of the incorrect dataset count. Furthermore, we would like to clarify that all images in the BCI and MIST datasets are real; the datasets themselves do not contain generated images, they are simply used as ground truth in the HE generation IHC task.
> >
> > **Comment 13:** Thank you to the author for checking. We have revised the caption.

---

> > > ### Author Response · Authors · 2026-01-25
> > >
> > > **Q1:** As shown in Comment 2, we present its distribution in the revised manuscript.
> > >
> > > **Q2:** We used the publicly available HISTAI database. We patched each WSI at 20x magnification, and each WSI also included a true-level image at 20x magnification, so no scaling was needed.
> > >
> > > **Q3&Q4:** Thank you for your question. We have provided an explanation in W4.
> > >
> > > **Q5:** 4870 is correct, and we have already made changes in the revised manuscript.
> > >
> > > **Q6:** Thank you for your question. We have provided an explanation in Comment 6.
> > >
> > > **Q7:** All PFMs employ the feature extraction process described in their respective official documentation. All feature extraction processes are available in their respective codebases.
> > >
> > > **Q8:** In our revised manuscript, we added larger-scale base models for Prov-GigaPath, H-Optimus-0, Virchow-2, and UNI-2, and compared them with StainNet-Small and StainNet-Base.
> > >
> > > **Q9:** Thank you for your question. We have provided an explanation in Comment 10.
> > >
> > > **Q10:** As shown in Comment 3, ablation experiments have been added for pre-trained data scales (25% and 50%) to evaluate their data scaling advantage.

---

### Official Review · Reviewer_iDGJ · 2026-01-10

**Confidence:** 4
**Preliminary Rating:** 3
**Final Rating:** 4

**Summary:**

The authors train a ViT-Small model using the DINO self-supervised method on 1.4 million tissue patches extracted from 20.2k special-stain WSIs, from the HistAI dataset.

They evaluate their trained model (StainNet) on one in-house WSI-level dataset and two public patch-level datasets, with special-stain images. The StainNet performance is compared to Resnet-Imagenet, CTransPath, UNI and a few other pathology foundation models.

**Strengths:**

- The studied problem is quite interesting, training a ViT model with DINO on special-stain WSIs. This is definitely relevant to the research community.

- Their trained StainNet model seems to perform quite well on special-stain images, matching the performance of UNI even though StainNet is a significantly smaller model (ViT-Small vs ViT-Large).

- Figure 6 is quite interesting, I agree with the authors that _"pre-training details such as the expected loss curve and convergence behavior are often underreported"_ within computational pathology.

**Weaknesses:**

- The paper could be a bit more well written overall.

- The evaluation could be more extensive. It includes a single WSI-level dataset, with less than 150 WSIs.

- The analysis could be more detailed to provide further insights. For example, evaluation of models trained on subsets of the dataset.

**Detailed Comments:**

Questions/suggestions:
- It would be interesting to also include an H&E dataset in the evaluation. StainNet matches the performance of UNI on special-stain images, but what about H&E? I think this could provide further insights on the benefits of special-stain training.
- It would be interesting to also train StainNet on e.g. 50% and 25% of the dataset, to check how the amount of training data affects the performance.
- If possible, it would be interesting to also train a ViT-Base version of StainNet and check how this affects the performance.
- Could it be possible to include one more WSI-level dataset in the evaluation?
- If you need more space in the main paper, you could show just one of the three metrics in Table 1 and move the full version to the appendix.
- In Table 1, it seems quite strange that Resnet-50 is better than the four other models for ABMIL (my experience is that pathology-specific models basically always outperform Resnet-Imagenet)?
- Is there a particular reason why you choose DINO and not DINOv2 (I'm mostly just curious)?
- Not sure that Figure 2 is interesting enough to be included in the main paper, could be moved to the appendix so make more space. Might also be more illustrative to show multiple augmented versions for each example instead.






Minor things:
- Abstract, "over 1.4 million patch images cropping from" --> "over 1.4 million patch images extracted from"?
- Abstract, "compare StainNet with recently larger PFMs" --> "compare StainNet with recent larger PFMs"?
- Section 1, "when H&E-stained resection or biopsy WSI are" --> "when H&E-stained resection or biopsy WSIs are".
- Section 4, "pre-trained on ImageNet datasets" --> "pre-trained on the ImageNet dataset"?
- Table 1 caption, "and report average balanced accuracy with standard deviation": you report two other metrics here as well.

**Justification Of Final Rating:**

The authors extended the experimental evaluation and analysis quite extensively. This is a quite interesting paper that definitely is relevant to the research community. While there is limited technical novelty, I still think this paper should be accepted.

**Justification Of The Preliminary Rating:**

I think this is a quite interesting paper that is relevant to the research community. However, I think it could be made significantly more interesting and relevant by extending the experimental evaluation and analysis. In its current state, I'm borderline on this paper.

**Questions To Address In The Rebuttal:**

Please see "Weaknesses" and "Questions/suggestions" above.

---

> ### Author Response · Authors · 2026-01-25
> **Response for Reviewer iDGJ**
>
> We appreciate the valuable comments from Reviewer iDGJ. We have made comprehensive revisions to the manuscript to address all concerns to the best of our ability. All changes in the revised manuscript are highlighted in **red**. Our responses are provided below.
>
> **W1:** Thank you for your careful review. We have carefully re-examined every section of the original manuscript and corrected previous typos and instances of poor phrasing in the revised manuscript. We hope the revised manuscript significantly improves the overall writing quality and meets the requirements of the MIDL conference.
>
> **W2:** We have added two WSI-level IHC tasks and three ROI-level special stain tasks to the revised manuscript to provide a more comprehensive evaluation of our StainNet models.
>
> **W3:** In the revised manuscript, we have included evaluations using different proportions of pre-training data. Moreover, we also use the same pre-training data to train a ViT-Base version (StainNet-Base) of our StainNet family to investigate the advantages across various model scales.
>
> **Q1:** We thank the reviewer for the insightful suggestion. Indeed, investigating the performance of models pre-trained on non-H\&E images when applied to H\&E images helps understand the uniqueness of IHC and special staining image features.
> We have added two ROI-level H\&E tasks: CRC-100K and KatherMS (evaluated via linear-probe fine-tuning), and one WSI-level H\&E task: PANDA-*Karo.* (evaluated via ABMIL fine-tuning) to conduct experiments.
>
> **Q2:** We thank the reviewer for this suggestion. To conduct an ablation study on the dataset scale, we performed random sampling at 25 and 50 patches per WSI to create pre-training subsets for training StainNet-Small. Results indicate that models trained on larger data scales consistently achieve superior performance. This ablation study has been added to the revised manuscript.
>
> **Q3:** We thank the reviewer for the valuable comments. First, we would like to apologize that due to limited computational resources before the original submission, we did not have sufficient time to train a larger-scale model. During the rebuttal phase, we expanded our computational capacity by using eight NVIDIA A100 (80GB) GPUs to train StainNet-Base with the ViT-Base architecture. All other pre-training hyperparameters remained identical to those used for StainNet-Small. The results demonstrate that StainNet-Base not only further improves the strength of the StainNet family but also achieves superior overall performance compared to several recent large-scale SOTA pathology foundation models, including UNI-2, Prov-GigaPath, Virchow-2, and H-optimus-0. We have added a detailed introduction to StainNet-Base and the corresponding experimental results in the revised manuscript.
>
> **Q4&Q5&Q8:** Thank you for your suggestion. We have added two new WSI-level tasks to the revised manuscript and also modified the manuscript format.
>
> **Q6:** We thank the reviewer for this insightful question. In fact, we were also surprised by these experimental results. We think this discrepancy may have started from the significant domain gap between non-H\&E and H\&E images. Specifically, IHC images primarily exhibit sparse, high-contrast cellular features, which are quite different from the dense texture and structural information emphasized in H\&E images. Therefore, more general models like ResNet-ImageNet, due to their lower dependence on specific H\&E-stained pathology domains, may be easier to generalize in certain situations. However, this does not imply that models pre-trained on natural images consistently outperform others on non-H&E tasks. For instance, in our newly added tasks involving glomerular Masson, PAS, and PASM stains, ResNet-ImageNet does not achieve the best performance.
>
> **Q7:** The main reason for choosing DINO rather than DINOv2 is its lower computational resource requirements. DINOv2 typically requires multi-node GPU servers to support training models with larger batch sizes and significantly more parameters. At the time of our original submission, our training was limited to a single PRO 9000 GPU, which did not meet the hardware requirements for the DINOv2 experiment. Fortunately, during the rebuttal phase, we gained access to the server with eight NVIDIA A100 (80GB) GPUs, which accelerated the training of StainNet-Small and enabled training the larger StainNet-Base version. We certainly believe that using DINOv2 with larger datasets and larger model scales would further enhance AI's representation capabilities for non-H&E-stained images. Although this will require substantial time and effort, we plan to explore this in our future work and firmly believe that further investigating the potential of special stains and IHC images is worthwhile.

---

> > ### Comment · Reviewer_iDGJ · 2026-01-31
> >
> > Thank you for the response, and for all the new experiments and analysis.
> >
> > I have increased my score.

---

### Author Rebuttal · Authors · 2026-01-25

**Rebuttal:**

We sincerely thank all reviewers for their time and effort in reviewing our manuscript. We have uploaded a revised version of the manuscript for the rebuttal. Although we have used different color codes to indicate our responses to each reviewer, we kindly invite the reviewers to also refer to responses in other colors for a more comprehensive evaluation of the revised manuscript. In addition, we would like to highlight the following two clarifications:

1. In the revised manuscript, we explicitly distinguish immunohistochemistry and special staining, which better aligns with standard pathology terminology and allows for a more precise description of non-H&E-stained images. We have also revised the paper title to further clarify this distinction.

2. To maintain naming consistency with the newly trained larger model, StainNet-Base, we have renamed the original StainNet to StainNet-Small in the revised manuscript.

**Supporting Material:**

/attachment/2b736d4f347780c24b0d4921fc7729fdb097c292.pdf

---

### Comment · Area_Chair_jBuR · 2026-01-28

Dear reviewers,

The authors have now responded to your reviews. At this time please participate in discussions with the authors.

IMPORTANT: You must enter your final rating by clicking “Edit” → “Official Review” and providing the Final Rating by February 1st 2026 (23:59 AoE).

Thank you again for your service to MIDL 2026 and making it a success.

---

### Comment · Area_Chair_jBuR · 2026-02-01
**please provide final ratings today**

Dear reviewers, if you have not done so already please provide your final ratings for the paper today. If you already did, please disregard this message.
Thanks!

---

### Meta-Review · Area_Chair_jBuR · 2026-02-05

**Recommendation:** Accept (Poster)
**Confidence:** 4

**Metareview:**

There were initially some concerns about the limited novelty the method and the narrow evaluation. After a detailed discussion between reviewers and authors, the evaluation weaknesses have been resolved. Two reviewers raised their scores from borderline to weak accept and weak accept to strong accept respectively after there rebuttal. One reviewer did not provide a final rating remaining at borderline. Overall, there is significant merit to this paper and it will be a valuable addition to MIDL.

---

### Decision · Program_Chairs · 2026-02-13

Accept (Poster)